# Multilingual Safety Alignment Via Sparse Weight Editing

**Jiaming Liang** [1]  **Zhaoxin Wang** [1]  **Handing Wang** [1]

## Abstract

Large Language Models (LLMs) exhibit significant safety disparities across languages, with low-resource languages (LRLs) often bypassing safety guardrails established for high-resource languages (HRLs) like English. Existing solutions, such as multilingual supervised fine-tuning (SFT) or Reinforcement Learning from Human Feedback (RLHF), are computationally expensive and dependent on scarce multilingual safety data. In this work, we propose a novel, training-free alignment framework based on *Sparse Weight Editing*. Identifying that safety capabilities are localized within a sparse set of "safety neurons", we formulate the cross-lingual alignment problem as a constrained linear transformation. We derive a closed-form solution to optimally map the harmful representations of LRLs to the robust safety subspaces of HRLs, while preserving general utility via a null-space projection constraint. Extensive experiments across 8 languages and multiple model families (Llama-3, Qwen-2.5) demonstrate that our method substantially reduces Attack Success Rate (ASR) in LRLs with negligible impact on general reasoning capabilities, all achieved with a single, data-efficient calculation. Our code is available at `Github`

## 1. Introduction

The rapid advancement of large language models (LLMs) has enabled impactful applications across domains(Achiam et al., 2023; Yang et al., 2025). However, when deployed in open and interactive environment, LLMs are exposed to diverse threats, raising safety concerns (Chander et al., 2025). For example, adversarial attacks (Szegedy et al., 2013; Goodfellow et al., 2014; Wang et al., 2024) can undermine reliability, while backdoor attacks can trigger malicious behaviors via data poisoning (Gu et al., 2019). Moreover, adversaries can exploit jailbreak attacks (Yi et al., 2024; Wang et al., 2025) to elicit harmful outputs.

To mitigate these risks, researchers have developed safety alignment techniques (Leike et al., 2018; Shao et al., 2026; Zhai et al., 2026; Kenton et al., 2021; Ji et al., 2023), including reinforcement learning from human feedback (RLHF) (Ouyang et al., 2022; Bai et al., 2022) and preference optimization methods (Rafailov et al., 2023; Shao et al., 2024), to align model behavior toward human values and social norms. Despite their effectiveness, these methods are data-intensive, requiring large-scale, carefully curated preference datasets, which are expensive and time-consuming to collect. This challenge is particularly acute in multilingual settings, as such datasets are abundant for high-resource languages (HRLs) like English but scarce for many low-resource languages (LRLs), leading to substantial cross-lingual disparities in safety. The same LLM is often well-aligned in English but considerably less safe in LRLs.

To bridge the gap of multilingual safety, the recent work (Bu et al., 2025; Zhao et al., 2025b;c) leverages multilingual corpora to improve safety in LRLs, often by relying on supervised fine-tuning (SFT) (Wei et al., 2021). However, these methods depend on costly, high-quality safety datasets in multiple languages. Some work (Xu et al., 2025a) transfers safety capabilities from HRLs to LRLs through intermediate languages bridge, but generally assumes strong translation performance. In practice, translation errors can propagate to downstream reasoning and generation, and translation-based pipelines introduce additional inference overhead.

Recent studies (Xu et al., 2025a) have identified the existence of "linguistic overlap neurons", specific neurons that are activated by both HRLs and LRLs and play a pivotal role in encoding core model capabilities, including safety mechanisms (Zhao et al., 2025d). This observation introduces a critical question:

### Can we transfer the safety representations of HRLs to LRLs without retraining?

In this work, we propose a multilingual alignment framework that transfers safety capabilities learned from HRLs (e.g., English) to LRLs. Concretely, we parameterize the cross-lingual adjustment as a low-rank transformation in

---
[*]Equal contribution  [1]School of Artificial Intelligence, Xidian University, Xi'an, China. Correspondence to: Handing Wang <hdwang@xidian.edu.cn>.

*Proceedings of the $43^{rd}$ International Conference on Machine Learning*, Seoul, South Korea. PMLR 306, 2026. Copyright 2026 by the author(s).

representation space, and solve a transformation matrix that maps the feature representations of harmful queries in LRLs to the well-aligned, safe activation patterns of HRLs. To ensure this modification does not compromise the model's utility, we introduce a null-space projection constraint derived from harmless data. This constraint ensures that our intervention is orthogonal to the directions encoding general capabilities, thereby modifying the safety-critical feature subspace, while minimizing effects on general capabilities.

Distinguishing our approach from prior work, we derive a closed-form solution to this optimization problem. This allows us to compute the optimal alignment parameters analytically using only a few anchor samples, eliminating the need for gradient-based training. Our contributions are summarized as follows:

- **Representation-Level Safety Transfer.** We introduce a representation alignment method for multilingual safety that maps the well-aligned and safe activation patterns from HRLs to LRLs tasks, thereby providing safety improvements across languages.

- **Training-Free Efficiency.** We formulate cross-lingual safety alignment as a regularized low-rank update problem and derive a closed-form solution. Our method requires only a small number of harmful and harmless anchor samples to compute the modification matrix, avoiding iterative gradient-based optimization.

- **Interpretable and Plug-and-Play Intervention.** Our framework provides an interpretable view of transferable safety-related representation components across languages. Moreover, it acts as a lightweight, plug-and-play intervention that can be integrated into different model architectures without disrupting parameters.

## 2. Related Works

### 2.1. Jailbreak Attacks

Despite their remarkable capabilities, LLMs remain vulnerable to adversarial exploitation. Attackers can craft *jailbreak* inputs—carefully engineered prompts designed to circumvent safety alignment and elicit harmful behaviors. Existing black-box jailbreak strategies primarily exploit the instruction-following nature of LLMs via sophisticated input manipulation and automated prompt search (Yi et al., 2024). Static approaches often leverage models' pattern-completion tendency by embedding malicious requests within benign-looking templates (Li et al., 2023; Yao et al., 2024; Anil et al., 2024; Wei et al., 2023). Such inputs may evade superficial safety filters while remaining interpretable to the underlying model. More advanced paradigms shift toward automated red-teaming, framing jailbreaking as an optimization problem. Using auxiliary LLMs together

with heuristics such as genetic algorithms or gradient-free optimization methods, these methods iteratively refine adversarial prompts to maximize attack success rate (Liu et al., 2024; Mehrotra et al., 2024; Chao et al., 2025).

### 2.2. Multilingual Safety Enhancement

**Training time.** Recent work extends safety alignment to multilingual settings by constructing cross-lingual safety datasets or leveraging HRLs signals as supervision. For example, AlignX (Bu et al., 2025) proposes a two-stage framework that first aligns multilingual representations and then fine-tunes the model with multilingual instructions to reduce the performance gap between HRLs and LRLs. Similarly, MPO (Zhao et al., 2025b) introduces a multilingual reward-gap optimization objective that minimizes discrepancies between reward distributions in HRLs (e.g., English) and LRLs, thereby facilitating cross-lingual safety transfer. AdaMergeX (Zhao et al., 2025c) explores cross-lingual transfer via adaptive adapter merging, aiming to decouple task competence from language competence.

**Inference time.** To circumvent the high costs of retraining, researchers have investigated inference-time interventions and parameter-efficient strategies. For example, RESTA (Bhardwaj et al., 2024) employs the task arithmetic to recover safety by adding a pre-computed safety vector—derived from the difference between an aligned and a deliberately unaligned model—to task-specific LLMs. However, this approach has several intrinsic drawbacks. First, the initial extraction of the safety vector necessitates a risky unalignment process and relies heavily on the coverage of the harmful datasets used. Second, the linear arithmetic operation on model weights lacks fine-grained control over specific linguistic neurons, often leading to a suboptimal trade-off between safety enforcement and the preservation of general capabilities.

**Translation-based methods.** Given the dominance of English-centric safety alignment, a widely used strategy is the Translate-Test pipeline (Ponti et al., 2021; Artetxe et al., 2023; Etxaniz et al., 2024), which translates LRLs inputs into English for safety processing. Extensions such as BridgeX-ICL (Xu et al., 2025a) further improve cross-lingual transfer by routing through bridge languages and exploiting linguistic overlap neurons. Despite their simplicity, translation-based methods face a fundamental semantic bottleneck that safety-critical intent may be altered or lost during translation, making safety enforcement in the original language less reliable.

### 2.3. Neuron Identification

Research on neuron-level interpretability has shifted from characterizing general model capabilities (Dai et al., 2022;

Wang et al., 2022) to identifying "Safety Neurons" critical for alignment. Current approaches typically locate these neurons through inference-time activation contrasting (Chen et al., 2026), linear probing classifiers (Wu et al., 2025), or ablation-based importance scoring (Zhao et al., 2025d). While findings on specific layer distribution vary, ranging from Feed-Forward Networks (Chen et al., 2026) to Self-Attention layers (Zhao et al., 2025d). These studies collectively establish that safety mechanisms are highly sparse, relying on less than 1% of total parameters to suppress harmful content effectively (Zhao et al., 2025d; Wu et al., 2025; Wang et al., 2026).

## 3. Empirical findings

### 3.1. Definition of Safety Neurons

Recent work in mechanistic interpretability suggests that important behaviors in large language models are often mediated by sparse and localized internal features or circuits rather than being uniformly distributed across the network (Dunefsky et al., 2024; Marks et al., 2024). In the safety setting, prior studies further show that safety-aligned behavior can be strongly affected by a relatively small subset of neurons or features, indicating that safety mechanisms in LLMs are at least partially localized (Chen et al., 2026; Zhao et al., 2025d; Wang et al., 2026).

Motivated by this line of work, we study whether such safety-relevant internal structure can support cross-lingual transfer. This question is especially relevant in multilingual LLMs, where internal representations are not uniformly shared across languages. Prior work reports stronger reliance on English-centric representations in multilingual reasoning (Etxaniz et al., 2024), while cross-lingual transfer in low-resource settings has also been linked to overlapping neuron patterns across languages (Xu et al., 2025b). Together, these findings suggest that multilingual models may contain safety-relevant components that are partially shared across languages, but not perfectly aligned.

Following activation-contrast-based safety-neuron identification methods (Chen et al., 2026; Wang et al., 2026), we define safety neurons as MLP units whose activations exhibit clear divergence between harmful and harmless inputs. Concretely, for each language $\ell$, we compare neuron activations elicited by harmful and harmless queries and select units that display both a large activation gap and strong statistical separability, yielding a language-specific safety-neuron set $S_\ell$. Full extraction details are provided in Appendix B.

**Assumption 3.1** (Sparse Safety Localization). Safety-related behavior can be effectively influenced through a sparse subset of neurons whose activations differ significantly between harmful and harmless inputs.

This formulation gives us a concrete handle for analyzing multilingual safety transfer. In Section 3.2, we test whether amplifying English safety neurons can causally influence safety behavior in other languages. In Section 3.3, we further examine when such transfer succeeds or fails by comparing the overlap of safety-neuron sets across languages.

### 3.2. Activation Steering for Cross-Lingual Safety

To verify whether the identified English safety neurons $\mathcal{S}_{eng}$ play a functional role in multilingual safety, we conduct an activation steering experiment. Our intuition is that if English acts as a dominant semantic anchor during training, strengthening English safety-related activations may improve safety behavior in other languages.

Specifically, during the forward pass of the model, we intervene on the activations of the identified safety neurons. For every neuron $j \in \mathcal{S}_{eng}$ at layer $l$, we scale its output activation $x_j^{(l)}$ by a coefficient $\alpha > 1$:

$$\tilde{x}_j^{(l)} = \alpha \cdot x_j^{(l)} \quad \forall j \in \mathcal{S}_{eng}, \tag{1}$$

where $\tilde{x}_j^{(l)}$ is the intervened activation. We evaluate the model's attack success rate (ASR) on multilingual jailbreak prompts under varying scaling factors $\alpha$.

As illustrated in Figure 1, simply amplifying English safety neurons significantly improves safety across various languages. This confirms that English safety neurons act as a universal safety neurons to some extent, leveraging the model's cross-lingual alignment.

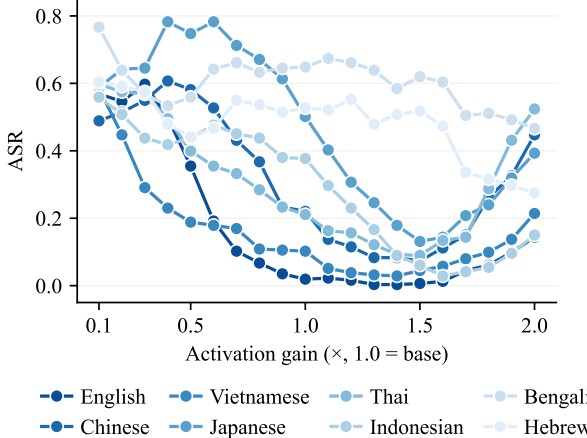

*Figure 1.* **Impact of English Safety Neuron Amplification.** Scaling the activations of English safety neurons leads to a consistent decrease in harmful response rates across multiple languages, validating the cross-lingual influence of these neurons.

### 3.3. Representation Transfer

While simple amplification can improve safety in some cases, we observe substantial variation in its effectiveness across languages. To further investigate the reason for this variability, we examine whether it is associated with **cross-lingual representation overlap**. We compute the **Jaccard similarity** (intersection over union) between the safety-neuron sets identified for each pair of languages. Formally, for languages $\ell_i$ and $\ell_j$, we define

$$\mathrm{Jaccard}(\mathcal{S}_{\ell_i}, \mathcal{S}_{\ell_j}) = \frac{|\mathcal{S}_{\ell_i} \cap \mathcal{S}_{\ell_j}|}{|\mathcal{S}_{\ell_i} \cup \mathcal{S}_{\ell_j}|}, \qquad (2)$$

where $\mathcal{S}_\ell$ denotes the safety-neuron index set extracted for language $\ell$. Figure 2 visualizes these pairwise similarities as a heatmap.

The heatmap reveals a clear overlap pattern that high-resource languages exhibit consistently higher *safety-neuron set* overlap, whereas low-resource languages show weaker overlap, both with high-resource languages and with one another. English has relatively high Jaccard similarity with several other languages, while many low-resource languages display more limited overlap and appear more isolated under this set-based similarity measure.

This observation explains the limitations of simple activation steering. When a target language already activates a safety-neuron subset aligned with the English-centric safety subspace, amplifying those neurons effectively suppresses harmful generation. However, for languages whose safety-relevant features are distributed over a distinct set of neurons, amplification primarily increases the magnitude of a mismatched activation pattern without correcting its direction. These findings highlight a fundamental geometric limitation of passive transfer mechanisms, that safety representations are not universally aligned across languages. As a result, effective multilingual safety alignment requires an *active reorientation* of language-specific safety representations toward a shared, robust safety anchor, rather than relying on incidental neuron overlap.

## 4. Method

Motivated by the empirical findings in Section 3, we propose SPARSE WEIGHT EDITING, a training-free alignment framework for bridging the representation gap between HRLs and LRLs. Our key observation is that the English safety subspace provides a reliable alignment anchor (Section 3.2), whereas many LRLs exhibit directional misalignment that makes simple activation steering ineffective (Section 3.3). We therefore cast cross-lingual safety transfer as a constrained linear transformation problem: we compute a sparse perturbation $\Delta W$ that aligns harmful representations in LRLs toward the safe activation patterns of HRLs, while

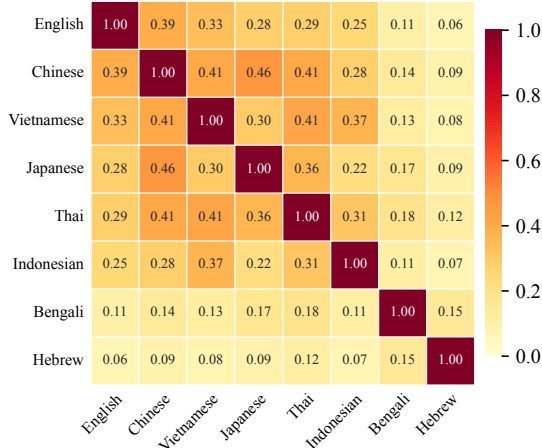

*Figure 2.* Pairwise safety-neuron set overlap across languages. Higher values indicate greater overlap under this set-based measure. HRLs tend to exhibit stronger overlap, whereas LRLs show weaker overlap both with HRLs and with each other.

preserving general utility via a null-space constraint.

### 4.1. Safety Neuron Identification

To identify the safety-critical neurons for each language, we construct a multilingual probing dataset by translating standard harmful ($\mathcal{D}_{harm}$) and harmless ($\mathcal{D}_{safe}$) corpora into our target languages (Appendix A). These language-specific probes enable us to contrast activations under harmful versus harmless inputs and localize the sparse neuronal subset most associated with safety behaviors, which we subsequently target in our weight editing procedure.

### 4.2. Cross-Lingual Safety Subspace Alignment

The empirical results in Section 3.3 reveal a fundamental limitation of direct safety transfer: *representation misalignment*. While HRLs such as English activate a distinct safety subspace, i.e., a characteristic activation pattern over safety neurons, LRL queries often induce feature representations that are orthogonal to or deviated from this subspace, likely due to insufficient safety supervision in the target language. As a result, simple activation amplification (Section 3.3) is ineffective for low-overlap languages. It increases the magnitude of an already misaligned representation without correcting its direction.

To address this, we explicitly *reorient* harmful representations in LRLs toward the safety pattern of HRLs via a weight-space linear mapping. Concretely, we solve for a sparse perturbation $\Delta W_{\mathcal{S}}$ applied to the safety weight submatrix $W_{\mathcal{S}}$ such that the projected activations for LRL harmful inputs $X_{low}$ match the target safety activations $Y_{target}$

derived from aligned HRLs:

$$\sigma(X_{low}(W_{\mathcal{S}} + \Delta W_{\mathcal{S}})) \approx Y_{target}. \qquad (3)$$

Here, $Y_{target}$ represents the desired safety activation pattern (e.g., the activations of English safety neurons under corresponding harmful queries). By minimizing the reconstruction error between the transformed LRL activations and $Y_{target}$, we enable cross-lingual safety transfer without retraining the full model.

### 4.3. Weight-Editing Formulation

We formulate cross-lingual safety transfer as a lightweight weight-editing problem on a small, safety-relevant subspace. The key idea is to (i) restrict the update to the identified safety neurons for parameter efficiency, (ii) align harmful LRL representations toward an English-derived safety activation target, and (iii) preserve benign utility via a null-space regularization. Finally, we impose a low-rank structure to improve robustness in the few-shot regime.

#### 4.3.1. SUBSPACE SELECTION

To minimize interference with general capabilities, we restrict weight editing strictly to the identified safety neurons. Let $\mathcal{S}$ denote the index set of safety neurons at layer $l$, with $|\mathcal{S}| = m$. The original weight matrix is $\boldsymbol{W} \in \mathbb{R}^{d_{in} \times d_{out}}$. We define the safety weight submatrix $\boldsymbol{W}_{\mathcal{S}} \in \mathbb{R}^{d_{in} \times m}$ as the columns of $\boldsymbol{W}$ indexed by $\mathcal{S}$. Our goal is to learn a perturbation $\Delta \boldsymbol{W}_{\mathcal{S}} \in \mathbb{R}^{d_{in} \times m}$ applied only to these columns, while keeping the remaining weights $\boldsymbol{W}_{\setminus \mathcal{S}}$ frozen.

#### 4.3.2. ALIGNMENT OBJECTIVE

We aim to align the harmful representations of LRLs with the safety activation patterns induced by English. Let $\boldsymbol{X}_{\text{low}} \in \mathbb{R}^{N_h \times d_{in}}$ denote the layer-$l$ input features extracted from harmful LRL queries, and let $\boldsymbol{Y}_{\text{target}} \in \mathbb{R}^{N_h \times m}$ denote the target activations of the safety neurons derived from the English context. We seek $\Delta \boldsymbol{W}_{\mathcal{S}}$ such that

$$\sigma(\boldsymbol{X}_{\text{low}}(\boldsymbol{W}_{\mathcal{S}} + \Delta \boldsymbol{W}_{\mathcal{S}})) \approx \boldsymbol{Y}_{\text{target}}. \qquad (4)$$

Directly optimizing Eq. 4 is inconvenient due to the nonlinearity $\sigma(\cdot)$. Following the motivation in Section 4.2, we adopt a first-order approximation in the pre-activation space and minimize the residual of the linear term:

$$\mathcal{L}_{\text{align}} = \|\boldsymbol{X}_{\text{low}} \Delta \boldsymbol{W}_{\mathcal{S}} - (\boldsymbol{Y}_{\text{target}} - \boldsymbol{X}_{\text{low}} \boldsymbol{W}_{\mathcal{S}})\|_F^2. \quad (5)$$

Define the *Safety Gap* as

$$\boldsymbol{D}_{\mathcal{S}} = \boldsymbol{Y}_{\text{target}} - \boldsymbol{X}_{\text{low}} \boldsymbol{W}_{\mathcal{S}} \in \mathbb{R}^{N_h \times m}, \qquad (6)$$

which captures the activation discrepancy that $\Delta \boldsymbol{W}_{\mathcal{S}}$ is expected to bridge.

#### 4.3.3. UTILITY CONSTRAINT AND REGULARIZATION

To preserve benign-task performance, we introduce a utility-preserving null-space regularization. Let $\boldsymbol{X}_{\text{safe}} \in \mathbb{R}^{N_s \times d_{in}}$ denote the layer-$l$ input features extracted from harmless queries. We encourage the perturbation $\Delta \boldsymbol{W}_{\mathcal{S}}$ to lie in the (right) null space of $\boldsymbol{X}_{\text{safe}}$, so that it induces minimal change on harmless features:

$$\mathcal{L}_{\text{utility}} = \|\boldsymbol{X}_{\text{safe}} \Delta \boldsymbol{W}_{\mathcal{S}}\|_F^2. \qquad (7)$$

#### 4.3.4. LOW-RANK CONSTRAINT

Optimizing a dense, full-rank perturbation $\Delta \boldsymbol{W}_{\mathcal{S}}$ is undesirable in our few-shot regime, where the anchor set is small relative to the number of free parameters. Without additional structure, a full-rank update can overfit to noise and exhibit poor generalization. Moreover, consistent with the *Sparse Safety Localization* assumption (Assumption 3.1), we expect the safety-relevant update to be concentrated in a low-dimensional subspace. We therefore impose a low-rank constraint $\text{rank}(\Delta \boldsymbol{W}_{\mathcal{S}}) \leq r$, which encourages the update to modify only the principal directions.

Combining the alignment objective, the utility regularization, and weight decay, the final optimization problem is:

$$\begin{aligned} \min_{\Delta \boldsymbol{W}_{\mathcal{S}}} \ & \|\boldsymbol{X}_{\text{low}} \Delta \boldsymbol{W}_{\mathcal{S}} - \boldsymbol{D}_{\mathcal{S}}\|_F^2 \quad + \gamma \|\boldsymbol{X}_{\text{safe}} \Delta \boldsymbol{W}_{\mathcal{S}}\|_F^2 \\ & + \lambda \|\Delta \boldsymbol{W}_{\mathcal{S}}\|_F^2 \\ \text{s.t.} \quad & \text{rank}(\Delta \boldsymbol{W}_{\mathcal{S}}) \leq r. \end{aligned} \qquad (8)$$

Since $\Delta \boldsymbol{W}_{\mathcal{S}} \in \mathbb{R}^{d_{in} \times m}$ only edits the columns corresponding to safety neurons, the update is lightweight compared to modifying the full weight matrix.

### 4.4. Closed-Form Solution

A key advantage of Eq. 8 is that it admits an analytic solution, avoiding iterative gradient-based optimization. In this section, we show that the rank constraint can be handled by reducing the objective to a standard low-rank approximation under a whitened metric, which yields an efficient single-pass solver.

**Theorem 4.1** (Low-Rank Safety Alignment). *Define*

$$\boldsymbol{Q} = \boldsymbol{X}_{low}^\top \boldsymbol{X}_{low} + \gamma \, \boldsymbol{X}_{safe}^\top \boldsymbol{X}_{safe} + \lambda \, \boldsymbol{I}. \qquad (9)$$

*For $\lambda > 0$, $\boldsymbol{Q}$ is positive definite and admits a Cholesky factorization $\boldsymbol{Q} = \boldsymbol{R}^\top \boldsymbol{R}$. Let*

$$\boldsymbol{M} = \boldsymbol{Q}^{-1} \boldsymbol{X}_{low}^\top \boldsymbol{D}_{\mathcal{S}} \qquad (10)$$

*be the optimal solution of Eq. 8 without the rank constraint. Then the optimal rank-$r$ perturbation $\Delta \boldsymbol{W}_{\mathcal{S}}^*$ for Eq. 8 is*

$$\Delta \boldsymbol{W}_{\mathcal{S}}^* = \boldsymbol{R}^{-1} \tilde{\boldsymbol{\Delta}}^*, \qquad (11)$$

*where $\tilde{\boldsymbol{\Delta}}^*$ is the best rank-$r$ approximation of $\tilde{\boldsymbol{M}} = \boldsymbol{RM}$ in Frobenius norm. Concretely, if $\tilde{\boldsymbol{M}} = \boldsymbol{U\Sigma V}^\top$ is the SVD of $\tilde{\boldsymbol{M}}$, then*

$$\tilde{\boldsymbol{\Delta}}^* = \boldsymbol{U\Sigma}_r \boldsymbol{V}^\top, \tag{12}$$

*with $\boldsymbol{\Sigma}_r$ keeping only the top-$r$ singular values (and setting the rest to zero).*

See Appendix C for the derivation based on the Eckart–Young–Mirsky theorem.

---

**Algorithm 1** Closed-Form Solver for Sparse Weight Editing

---

1: **Input:** $\boldsymbol{X}_{\text{low}}, \boldsymbol{X}_{\text{safe}}, \boldsymbol{W}_{\mathcal{S}}, \boldsymbol{Y}_{\text{target}}, \gamma, \lambda, r$
2: **Output:** $\Delta \boldsymbol{W}_{\mathcal{S}}^*$
3: /* compute safety gap */
4: $\boldsymbol{D}_{\mathcal{S}} \leftarrow \boldsymbol{Y}_{\text{target}} - \boldsymbol{X}_{\text{low}} \boldsymbol{W}_{\mathcal{S}}$
5: /* build metric matrix and whiten */
6: $\boldsymbol{Q} \leftarrow \boldsymbol{X}_{\text{low}}^\top \boldsymbol{X}_{\text{low}} + \gamma \boldsymbol{X}_{\text{safe}}^\top \boldsymbol{X}_{\text{safe}} + \lambda \boldsymbol{I}$
7: Compute Cholesky factorization $\boldsymbol{Q} = \boldsymbol{R}^\top \boldsymbol{R}$
8: /* compute unconstrained ridge solution */
9: $\boldsymbol{M} \leftarrow \boldsymbol{Q}^{-1} \boldsymbol{X}_{\text{low}}^\top \boldsymbol{D}_{\mathcal{S}}$
10: $\tilde{\boldsymbol{M}} \leftarrow \boldsymbol{RM}$
11: /* rank-$r$ approximation in whitened space */
12: Compute truncated SVD $\tilde{\boldsymbol{M}} \approx \boldsymbol{U\Sigma}_r \boldsymbol{V}^\top$
13: /* unwhiten to obtain the final update */
14: $\Delta \boldsymbol{W}_{\mathcal{S}}^* \leftarrow \boldsymbol{R}^{-1} \left( \boldsymbol{U\Sigma}_r \boldsymbol{V}^\top \right)$
15: **Return** $\Delta \boldsymbol{W}_{\mathcal{S}}^*$

---

**Practical computation.** The closed-form update can be computed in one pass via Cholesky solves and a rank-$r$ truncated SVD on $\tilde{\boldsymbol{M}} \in \mathbb{R}^{d_{in} \times m}$; see Algorithm 1.

## 5. Experiments

We evaluate whether our lightweight alignment update $\Delta \boldsymbol{W}$ (i) consistently reduces harmful completions under multilingual jailbreak prompts and (ii) preserves general capabilities across languages under a strict zero-shot protocol. We further examine the compatibility of our method with an existing safety-alignment baseline MPO(Zhao et al., 2025b) and provide ablations on key design choices.

### 5.1. Experimental Setup

#### 5.1.1. DATASETS

To ensure a strict zero-shot evaluation, we use disjoint datasets for the alignment phase (computing $\Delta \boldsymbol{W}$) and the evaluation phase.

**Evaluation benchmark ($\mathcal{D}_{test}$).** We construct a multilingual safety benchmark, MULTI-STRONGREJECT, by trans-

lating the English `walledai/StrongREJECT` (Souly et al., 2024) benchmark into seven additional languages using `tencent/Hunyuan-MT-7B` (Zheng et al., 2025). MULTI-STRONGREJECT covers eight languages: English (En), Chinese (Zh), Vietnamese (Vi), Japanese (Ja), Thai (Th), Indonesian (Id), Bengali (Bn), and Hebrew (He), spanning diverse language families and resource levels. Each language subset contains 313 harmful queries designed to probe safety vulnerabilities.

#### 5.1.2. MODELS

We evaluate our approach on representative LLMs families across multiple parameter scales to assess cross-model robustness. Specifically, we consider Llama-3.2, Qwen2 and Qwen2.5 from 1B to 7B. These models cover a range of sizes and pretraining corpora, providing a broad testbed for evaluating generality.

#### 5.1.3. EVALUATION METRICS

**Safety.** We report *Attack Success Rate (ASR)* as the primary safety metric. For scalable multilingual evaluation, we use `Qwen/Qwen3Guard-Gen-8B` (Zhao et al., 2025a) to classify response harmfulness. A query is counted as an attack success if the guard model flags the generated response as unsafe. To assess whether the observed trends depend on this particular evaluator, we further conduct a second-judge evaluation with GPTJudge (GPT-5) in Appendix F.3.

**Utility.** To quantify the safety and utility trade-off, we evaluate: **MGSM** (Multilingual Grade School Math) for cross-lingual reasoning, and **M-MMLU** (Multilingual Massive Multitask Language Understanding) for multilingual general knowledge. We report the average accuracy across the target languages as an overall utility summary.

### 5.2. Main Results

Table 1 shows that applying our lightweight update $\Delta \boldsymbol{W}$ consistently reduces harmful completions across model families and languages under a strict zero-shot protocol, where translated evaluation prompts are never observed during alignment. This demonstrates that our method does not rely on language-specific supervision at test time, but instead induces a transferable safety adjustment in the model's internal representations.

The safety gains are particularly pronounced for low-resource languages and smaller backbones (e.g., Qwen2-0.5B and Qwen2-1.5B), where the unaligned models exhibit high attack success rates. In these settings, $\Delta \boldsymbol{W}$ yields substantial absolute reductions in unsafe responses, suggesting that our approach effectively corrects representation-level misalignment that disproportionately affects under-resourced languages. By contrast, for larger or already

*Table 1.* **Zero-shot multilingual safety and utility evaluation.** Safety is reported as the number of unsafe responses flagged by `Qwen3Guard-Gen-8B` out of 313 prompts (lower is better). Superscripts denote the change in unsafe-response counts compared to **None** for the same backbone (negative indicates improvement; positive indicates regression). $\Delta_{Avg}$ denotes the average change across the reported languages. Utility is measured by MGSM and M-MMLU accuracy (higher is better).

| MODELS | METHOD | SAFETY | | | | | | | UTILITY | |
|---|---|---|---|---|---|---|---|---|---|---|
| | | ASR ↓ (#UNSAFE / 313) | | | | | | | MGSM↑ | M-MMLU↑ |
| | | EN | ZH | VI | JA | BN | HE | $\Delta_{Avg}$ | | |
| LLAMA-3.2-1B | NONE | 6/313 | 61/313 | 31/313 | 149/313 | 179/313 | 109/313 | - | 18.58 | 26.54 |
| | OUR | 0/313$^{-6}$ | 27/313$^{-34}$ | 4/313$^{-27}$ | 81/313$^{-68}$ | 144/313$^{-35}$ | 115/313$^{+6}$ | -27.33 | 18.36 | 27.22 |
| | MPO | 0/313$^{-6}$ | 22/313$^{-39}$ | 9/313$^{-22}$ | 78/313$^{-71}$ | 152/313$^{-27}$ | 135/313$^{+26}$ | -23.15 | 19.64 | 25.96 |
| | MPO+OUR | 0/313$^{-6}$ | 22/313$^{-39}$ | 0/313$^{-31}$ | 66/313$^{-83}$ | 96/313$^{-83}$ | 109/313$^{-0}$ | -40.33 | 19.45 | 26.58 |
| LLAMA-3.2-3B | NONE | 6/313 | 9/313 | 10/313 | 79/313 | 110/313 | 39/313 | - | 32.76 | 37.10 |
| | OUR | 4/313$^{-2}$ | 3/313$^{-6}$ | 2/313$^{-8}$ | 34/313$^{-45}$ | 65/313$^{-45}$ | 46/313$^{+7}$ | -16.5 | 32.76 | 37.00 |
| | MPO | 4/313$^{-2}$ | 8/313$^{-1}$ | 4/313$^{-6}$ | 50/313$^{-29}$ | 91/313$^{-19}$ | 36/313$^{-3}$ | -10.0 | 33.67 | 36.88 |
| | MPO+OUR | 2/313$^{-4}$ | 1/313$^{-8}$ | 3/313$^{-7}$ | 30/313$^{-49}$ | 58/313$^{-52}$ | 36/313$^{-3}$ | -20.5 | 32.76 | 36.76 |
| QWEN2-0.5B | NONE | 224/313 | 197/313 | 185/313 | 193/313 | 208/313 | 150/313 | - | 7.75 | 32.71 |
| | OUR | 176/313$^{-48}$ | 121/313$^{-76}$ | 139/313$^{-46}$ | 145/313$^{-48}$ | 173/313$^{-35}$ | 134/313$^{-16}$ | -44.83 | 5.27 | 31.01 |
| | MPO | 108/313$^{-116}$ | 93/313$^{-104}$ | 83/313$^{-102}$ | 94/313$^{-99}$ | 162/313$^{-46}$ | 90/313$^{-60}$ | -87.83 | 4.80 | 32.69 |
| | MPO+OUR | 56/313$^{-168}$ | 41/313$^{-156}$ | 44/313$^{-141}$ | 49/313$^{-144}$ | 120/313$^{-88}$ | 65/313$^{-85}$ | -130.33 | 4.36 | 32.39 |
| QWEN2-1.5B | NONE | 36/313 | 18/313 | 36/313 | 67/313 | 187/313 | 83/313 | - | 20.95 | 41.63 |
| | OUR | 5/313$^{-31}$ | 4/313$^{-14}$ | 15/313$^{-21}$ | 19/313$^{-48}$ | 150/313$^{-37}$ | 36/313$^{-47}$ | -33 | 20.33 | 41.58 |
| | MPO | 0/313$^{-36}$ | 2/313$^{-16}$ | 0/313$^{-36}$ | 3/313$^{-64}$ | 21/313$^{-166}$ | 3/313$^{-80}$ | -66.33 | 19.38 | 41.44 |
| | MPO+OUR | 3/313$^{-33}$ | 0/313$^{-18}$ | 5/313$^{-31}$ | 1/313$^{-66}$ | 1/313$^{-186}$ | 1/313$^{-82}$ | -69.33 | 18.22 | 41.39 |
| QWEN2.5-1.5B | NONE | 60/313 | 30/313 | 42/313 | 56/313 | 182/313 | 118/313 | - | 27.53 | 41.58 |
| | OUR | 17/313$^{-43}$ | 5/313$^{-25}$ | 14/313$^{-28}$ | 14/313$^{-42}$ | 152/313$^{-30}$ | 81/313$^{-37}$ | -34.16 | 25.13 | 41.89 |
| | MPO | 6/313$^{-54}$ | 2/313$^{-28}$ | 1/313$^{-41}$ | 2/313$^{-54}$ | 54/313$^{-128}$ | 26/313$^{-92}$ | -62.66 | 23.09 | 40.78 |
| | MPO+OUR | 5/313$^{-55}$ | 2/313$^{-28}$ | 2/313$^{-40}$ | 7/313$^{-49}$ | 56/313$^{-126}$ | 22/313$^{-96}$ | -65.66 | 22.29 | 40.73 |
| QWEN2.5-3B | NONE | 61/313 | 64/313 | 64/313 | 81/313 | 157/313 | 100/313 | - | 31.02 | 47.18 |
| | OUR | 14/313$^{-47}$ | 4/313$^{-60}$ | 7/313$^{-57}$ | 15/313$^{-66}$ | 112/313$^{-45}$ | 41/313$^{-59}$ | -55.66 | 30.91 | 44.87 |
| | MPO | 16/313$^{-45}$ | 10/313$^{-54}$ | 10/313$^{-54}$ | 16/313$^{-65}$ | 67/313$^{-90}$ | 32/313$^{-68}$ | -62.66 | 36.62 | 46.14 |
| | MPO+OUR | 6/313$^{-55}$ | 5/313$^{-59}$ | 3/313$^{-61}$ | 4/313$^{-77}$ | 25/313$^{-131}$ | 7/313$^{-93}$ | -79.5 | 36.00 | 47.05 |
| QWEN2.5-7B | NONE | 16/313 | 12/313 | 21/313 | 39/313 | 98/313 | 48/313 | - | 32.00 | 49.37 |
| | OUR | 3/313$^{-13}$ | 5/313$^{-7}$ | 6/313$^{-15}$ | 9/313$^{-30}$ | 60/313$^{-38}$ | 24/313$^{-24}$ | -21.16 | 31.56 | 49.19 |
| | MPO | 6/313$^{-10}$ | 5/313$^{-7}$ | 5/313$^{-16}$ | 8/313$^{-31}$ | 25/313$^{-73}$ | 17/313$^{-31}$ | -28.0 | 38.36 | 47.16 |
| | MPO+OUR | 0/313$^{-16}$ | 0/313$^{-12}$ | 1/313$^{-20}$ | 2/313$^{-37}$ | 11/313$^{-87}$ | 11/313$^{-37}$ | -34.83 | 38.65 | 47.72 |

better-aligned models, improvements are more moderate but remain consistent, indicating that the update adapts to different baseline safety levels rather than overfitting to a specific regime. For readability, Table 1 reports a representative subset of languages; complete results over all languages are included in Appendix F. We further evaluate Swahili and Javanese as additional lower-resource languages in Appendix F.4, provide additional results on MULTIJAIL in Appendix F.1, and compare with MCD in Appendix F.2.

Our method is also highly compatible with existing safety alignment techniques. Across nearly all evaluated backbones, combining our update with MPO (**MPO+Our**) achieves the lowest unsafe-response counts, demonstrating that our training-free weight edit acts as a complementary safety plug-in rather than a replacement for existing methods.

Importantly, the improved safety does not come at the expense of general capabilities. Performance on MGSM and M-MMLU remains close to the **None** baseline in most cases, with only minor fluctuations across models and languages. In several settings, **MPO+Our** even matches or exceeds MPO in utility at comparable or stronger safety levels. These results indicate that our lightweight, training-

free update can improve multilingual safety while largely preserving general reasoning and knowledge, supporting its practicality as a drop-in safety enhancement.

### 5.3. Ablation Study

We conduct ablations on `Llama-3.2-1B` to isolate the impact of three key components in SPARSE WEIGHT EDITING: the safety neuron identification method, anchor construction for the utility constraint, and the rank $r$ of the low-rank update.

**Safety neuron identification method.** We further ablate the effect of the safety neuron identification strategy. Besides our proposed extraction procedure, we consider an alternative probe-based method adopted in NEUROSTRIKE (Wu et al., 2025). Concretely, NeuroStrike trains a safety probe (a lightweight linear classifier) on activation-label pairs to predict whether an input is harmful. It then selects safety neurons by ranking probe weights: neurons with large-magnitude *positive* weights (after z-score normalization) are treated as safety-critical dimensions. In this ablation, we replace our safety-neuron set with the probe-selected neurons from NeuroStrike, while keeping the rest of

our training-free alignment pipeline unchanged. We denote this variant as **Probe-selected**. Table 2 shows that the probe-selected neurons also lead to a substantial ASR reduction compared with the unedited baseline, suggesting that our editing formulation can be instantiated with different safety-neuron selectors. Moreover, combining the probe-selected variant with MPO further improves safety while maintaining utility, indicating that the effect is complementary to MPO.

*Table 2.* **Ablation on safety-neuron selection.** We replace our safety-neuron extraction with a probe-based selector inspired by NEUROSTRIKE (**Probe-selected**), while keeping the subsequent sparse weight-editing stage unchanged. MPO is included to show compatibility with an existing alignment method. We report safety (ASR; lower is better) and utility (MGSM, M-MMLU; higher is better).

| Method | ASR↓ | MGSM↑ | M-MMLU↑ |
|---|---|---|---|
| None | 28.27 | 18.58 | 26.54 |
| Probe-selected | 14.93 | 17.71 | 27.14 |
| MPO | 19.52 | 19.64 | 25.96 |
| MPO + Probe-selected | 12.53 | 19.53 | 26.54 |

**Anchor selection.** We first examine the effect of anchor data selection, which directly relates to the null-space utility constraint in our formulation. Table 3 compares three variants: using both UtilityAnchor and Regular, using UtilityAnchor alone, and using Regular alone.

Using both UtilityAnchor and Regular achieves the best overall safety–utility trade-off. Although UtilityAnchor alone substantially alters the solution, it leads to pronounced utility degradation (MGSM drops to nearly zero) and weak safety performance. This indicates that optimizing against UtilityAnchor alone biases the update toward preserving benign behavior while failing to sufficiently correct harmful behavior. Conversely, using Regular alone better preserves utility but yields weaker safety gains. Overall, these results demonstrate that balanced anchor construction is essential for preventing over-alignment while maintaining strong safety improvements.

*Table 3.* **Anchor choice ablation on `Llama-3.2-1B`.** We vary whether the alignment uses UtilityAnchor and/or Regular and report safety (ASR; lower is better) and utility (MGSM, M-MMLU; higher is better).

| CHOICE ↓ / MODELS → | LLAMA-3.2-1B | | |
|---|---|---|---|
| UTILITYANCHOR | ✓ | ✓ | |
| REGULAR | ✓ | | ✓ |
| ASR ↓ | 17.53 | 68.57 | 17.25 |
| MGSM ↑ | 18.36 | 0.11 | 11.02 |
| M-MMLU ↑ | 27.22 | 24.21 | 26.02 |

**Effect of rank $r$.** We next analyze the sensitivity of our method to the rank constraint $r$, which encodes the low-dimensional structure assumption underlying SPARSE WEIGHT EDITING. We vary $r$ from 4 to 512 while keeping all other settings fixed.

As shown in Table 4, ASR quickly saturates and remains stable across a wide range of ranks. Notably, small ranks (e.g., $r = 8$ or 16) already achieve safety performance comparable to much larger ranks. At the same time, utility metrics (MGSM and M-MMLU) are nearly invariant to the choice of $r$.

These results provide empirical support for our low-rank design: the transferable safety update resides in a low intrinsic-dimensional subspace, where the leading singular directions of $\tilde{M}$ capture most of the safety-relevant signal. From a practical perspective, this robustness indicates that our method does not rely on careful tuning of $r$, and low-rank settings suffice to obtain strong and stable safety gains.

*Table 4.* **Effect of rank $r$ on safety and utility.** Results are reported on `Llama-3.2-1B`. Performance remains stable across a wide range of ranks, indicating low sensitivity to the rank choice.

| RANK $r$ | ASR ↓ (%) | MGSM ↑ (%) | M-MMLU ↑ (%) |
|---|---|---|---|
| 4 | 15.42 | 18.33 | 27.19 |
| 8 | 15.54 | 17.96 | 27.19 |
| 16 | 15.34 | 18.18 | 27.18 |
| 32 | 17.53 | 18.36 | 27.22 |
| 64 | **14.90** | 18.29 | 27.22 |
| 128 | 14.98 | 18.11 | 27.19 |
| 256 | 16.41 | 18.11 | 27.18 |
| 512 | 16.17 | 18.11 | 27.19 |

## 6. Conclusion

We presented SPARSE WEIGHT EDITING, a training-free alignment framework for cross-lingual safety transfer. Motivated by the observation that low-resource languages often exhibit representation misalignment with the English safety subspace, we cast multilingual safety alignment as a constrained weight-space mapping problem over a small set of safety neurons. Our method computes a sparse, low-rank perturbation $\Delta W$ that reorients harmful LRL representations toward an English-derived safety activation target, while preserving benign utility via a null-space regularization. The resulting objective admits a closed-form solution, enabling efficient one-pass updates without gradient-based fine-tuning.

Across multiple model families and languages, experiments on MULTI-STRONGREJECT show that our training-free update consistently reduces harmful completions under a strict zero-shot protocol, and can be deployed as a lightweight post-hoc plug-in that composes with MPO to deliver additional safety gains. Importantly, these improvements

typically incur limited utility regression on MGSM and M-MMLU, suggesting that targeted subspace editing can improve safety without catastrophically degrading general capabilities. Ablations further highlight that balanced anchor construction is crucial for avoiding over-alignment while maintaining strong safety improvements.

Our work opens several directions for future research. First, developing more principled anchor selection strategies and automatically adapting hyperparameters (e.g., rank and regularization strengths) could further improve robustness across backbones and languages. Second, extending sparse weight editing beyond a single layer or neuron subset to multi-layer, hierarchical safety subspaces may provide stronger guarantees against adaptive jailbreaks. Finally, integrating our framework with stronger multilingual evaluators and more diverse safety taxonomies could help characterize when and why safety directions transfer across languages, enabling more reliable multilingual alignment in practice.

## Acknowledgements

This work was supported in part by the National Natural Science Foundation of China (No. 62376202).

## Impact Statement

This paper presents work whose goal is to advance the field of Machine Learning. There are many potential societal consequences of our work, none which we feel must be specifically highlighted here.

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

# A. Details of Multilingual Dataset Construction

We construct our multilingual corpus via a translation-based pipeline. Starting from an English seed set, we use `tencent/Hunyuan-MT-7B` (Zheng et al., 2025) to translate each example into the eight target languages (En, Zh, Vi, Ja, Th, Id, Bn, He). This procedure is applied consistently to both harmful and harmless subsets, producing language-parallel counterparts that enable controlled probing and alignment while keeping the underlying intent distribution fixed across languages.

| Subset | Source datasets | Description |
|--------|-----------------|-------------|
| Harmful ($\mathcal{D}_{harm}$) | HarmfulQA, CatHarmfulQA, LLM-LAT | Queries spanning diverse malicious and unsafe intents |
| Harmless ($\mathcal{D}_{safe}$) | NaturalReasoning | Benign queries used as control samples |

*Table 5.* Composition of the English seed set prior to translation.

## A.1. Translation Robustness Checks

We use `tencent/Hunyuan-MT-7B` for translation because it can be deployed locally and supports high-throughput generation, which makes large-scale multilingual evaluation reproducible without relying on rate-limited commercial APIs. Since machine translation may weaken or distort harmful intent, we further conduct two checks. First, we re-translate Swahili and Javanese prompts with Google Translate and re-run the evaluation; the results remain consistent with the main trend in Table 13. Second, we perform a back-translation semantic-preservation check over 2,817 prompts across nine translated languages. The mean BERT cosine similarity between the original English prompts and the back-translated prompts is 0.939, and only 57 prompts (2.02%) show noticeable semantic drift. These results suggest that translation noise is limited in scope, although translation-based benchmarks still cannot fully replace native-speaker safety evaluations.

*Table 6.* **Back-translation semantic preservation.** We report the mean BERT cosine similarity between the original English prompts and their back-translated English versions. Higher indicates stronger semantic preservation.

| LANGUAGE | ZH | VI | JA | TH | ID | BN | HE | SW | JV |
|----------|----|----|----|----|----|----|----|----|----|
| SIMILARITY | 0.93 | 0.93 | 0.92 | 0.93 | 0.94 | 0.94 | 0.92 | 0.93 | 0.95 |

# B. Details of Safety Neuron Extraction

We adopt a dual-metric extraction procedure following prior practice, including a concurrent submission by the authors (Wang et al., 2026). **This extraction is used solely to instantiate the sparse unit set required by Assumption 3.1 and is not the primary contribution of this work.**

In this section, we provide the mathematical formulation and implementation details for identifying safety neurons. Our goal is to isolate the sparse subset of neurons within the MLP blocks (specifically the `up_proj` and `gate_proj` weights) that exhibit significant activation divergence when processing harmful versus harmless inputs.

## B.1. Data Collection

Let $\mathcal{D}_{harm}$ and $\mathcal{D}_{safe}$ denote the datasets containing $N$ harmful and $N$ harmless queries, respectively. We feed these inputs into the model and record the activations of the MLP neurons. For a specific layer $l$ and neuron $j$, let $A_{l,j}^{(harm)}$ and $A_{l,j}^{(safe)}$ represent the sets of scalar activation values collected from the respective datasets. We compute the sample means $\bar{A}_{l,j}^{(harm)}$ and $\bar{A}_{l,j}^{(safe)}$ to represent the neuron's average response intensity.

## B.2. Selection Criteria 1: Activation Magnitude Difference

This criterion identifies neurons that act as primary triggers, showing a sharp intensity increase for harmful content. To assess the significance of a neuron's response relative to the entire layer, we employ z-score standardization on the activation differences.

First, we calculate the raw activation difference for every neuron $j$ in layer $l$:

$$\Delta_{l,j} = \bar{A}_{l,j}^{(harm)} - \bar{A}_{l,j}^{(safe)}$$

Next, we compute the mean ($\mu_\Delta^{(l)}$) and standard deviation ($\sigma_\Delta^{(l)}$) of these difference values across all neurons in the layer:

$$\mu_\Delta^{(l)} = \mathop{\mathbb{E}}_{j \in \text{Layer } l}[\Delta_{l,j}], \quad \sigma_\Delta^{(l)} = \sqrt{\mathop{\mathbb{E}}_{j \in \text{Layer } l}[(\Delta_{l,j} - \mu_\Delta^{(l)})^2]}$$

We then define the z-score for neuron $j$ as:

$$z_{l,j} = \frac{\Delta_{l,j} - \mu_\Delta^{(l)}}{\sigma_\Delta^{(l)}}$$

**Definition B.1** (Magnitude-based Candidate Set). We select neurons whose activation difference is statistically significant, i.e., it deviates from the layer's average behavior by more than $\tau_{mag}$ standard deviations:

$$\mathcal{S}_{mag}^{(l)} = \{j \mid z_{l,j} > \tau_{mag}\} \tag{13}$$

In our experiments, we set $\tau_{mag} = 2.0$, effectively selecting the outliers that are highly sensitive to harmful features.

### B.3. Selection Criteria 2: Statistical Effect Size (Cohen's $d$)

Solely relying on mean differences can be susceptible to outliers (e.g., a neuron that activates extremely highly for only a single harmful sample). To ensure the separation between harmful and harmless distributions is consistent, we employ Cohen's $d$.

**Definition B.2** (Significance-based Candidate Set). The Cohen's $d$ value for neuron $j$ is calculated as:

$$d_{l,j} = \frac{\bar{A}_{l,j}^{(harm)} - \bar{A}_{l,j}^{(safe)}}{s_{pooled}} \tag{14}$$

where $s_{pooled}$ is the pooled standard deviation of the two sample sets. We define the candidate set as:

$$\mathcal{S}_{stat}^{(l)} = \{j \mid d_{l,j} > \tau_{stat}\} \tag{15}$$

where $\tau_{stat}$ is empirically set to 1.0. A high $d_{l,j}$ indicates a robust distributional separation.

### B.4. Final Safety Neuron Aggregation

The final set of safety neurons for layer $l$ is the union of the two candidate sets:

$$\mathcal{S}_{safety}^{(l)} = \mathcal{S}_{mag}^{(l)} \cup \mathcal{S}_{stat}^{(l)} \tag{16}$$

This strategy ensures robust identification by capturing both high-intensity triggers and reliable discriminators.

## C. Proof of Theorem 4.1

We prove Theorem 4.1 by reducing Eq. 8 to a standard low-rank approximation problem.

**Problem.** Recall the rank-constrained objective:

$$\min_{\Delta \boldsymbol{W}_\mathcal{S}:\ \text{rank}(\Delta \boldsymbol{W}_\mathcal{S}) \leq r} \mathcal{J}(\Delta \boldsymbol{W}_\mathcal{S}), \tag{17}$$

where

$$\mathcal{J}(\Delta \boldsymbol{W}_\mathcal{S}) = \|\boldsymbol{X}_{\text{low}} \Delta \boldsymbol{W}_\mathcal{S} - \boldsymbol{D}_\mathcal{S}\|_F^2 + \gamma \|\boldsymbol{X}_{\text{safe}} \Delta \boldsymbol{W}_\mathcal{S}\|_F^2 + \lambda \|\Delta \boldsymbol{W}_\mathcal{S}\|_F^2. \tag{18}$$

**Step 1: Quadratic form and completion of the square.** Expanding Eq. 18 and collecting terms that depend on $\Delta W_{\mathcal{S}}$ yields

$$\mathcal{J}(\Delta W_{\mathcal{S}}) = \text{Tr}\left(\Delta W_{\mathcal{S}}^{\top} Q \Delta W_{\mathcal{S}}\right) - 2\,\text{Tr}\left(D_{\mathcal{S}}^{\top} X_{\text{low}} \Delta W_{\mathcal{S}}\right) + \text{Tr}\left(D_{\mathcal{S}}^{\top} D_{\mathcal{S}}\right), \tag{19}$$

with

$$Q = X_{\text{low}}^{\top} X_{\text{low}} + \gamma\, X_{\text{safe}}^{\top} X_{\text{safe}} + \lambda\, I. \tag{20}$$

For $\lambda > 0$, $Q$ is symmetric positive definite. Define the $Q$-weighted norm $\|Z\|_{Q}^{2} \triangleq \text{Tr}(Z^{\top} Q Z)$. Let

$$M = Q^{-1} X_{\text{low}}^{\top} D_{\mathcal{S}}. \tag{21}$$

Then Eq. 19 can be written as

$$\mathcal{J}(\Delta W_{\mathcal{S}}) = \|\Delta W_{\mathcal{S}} - M\|_{Q}^{2} + \text{const}, \tag{22}$$

where const does not depend on $\Delta W_{\mathcal{S}}$. Therefore, the original problem is equivalent to

$$\min_{\Delta W_{\mathcal{S}}:\ \text{rank}(\Delta W_{\mathcal{S}}) \leq r} \|\Delta W_{\mathcal{S}} - M\|_{Q}^{2}. \tag{23}$$

**Step 2: Whitening via Cholesky factorization.** Since $Q \succ 0$, let $Q = R^{\top} R$ be its Cholesky factorization with invertible $R$. Then

$$\|\Delta W_{\mathcal{S}} - M\|_{Q}^{2} = \|R\left(\Delta W_{\mathcal{S}} - M\right)\|_{F}^{2}. \tag{24}$$

Define $\tilde{\Delta} \triangleq R \Delta W_{\mathcal{S}}$ and $\tilde{M} \triangleq R M$. Because $R$ is invertible, left-multiplication preserves rank, i.e., $\text{rank}(\tilde{\Delta}) = \text{rank}(\Delta W_{\mathcal{S}})$. Thus Eq. 23 becomes

$$\min_{\tilde{\Delta}:\ \text{rank}(\tilde{\Delta}) \leq r} \left\|\tilde{\Delta} - \tilde{M}\right\|_{F}^{2}. \tag{25}$$

**Step 3: Optimal rank-$r$ approximation.** By the Eckart–Young–Mirsky theorem, the minimizer of Eq. 25 is given by the rank-$r$ truncated SVD of $\tilde{M}$. Let $\tilde{M} = U \Sigma V^{\top}$ be its SVD; then

$$\tilde{\Delta}^{*} = U \Sigma_{r} V^{\top}, \tag{26}$$

where $\Sigma_{r}$ keeps only the top-$r$ singular values (others set to zero).

**Step 4: Recovering $\Delta W_{\mathcal{S}}^{*}$.** Finally, mapping back yields

$$\Delta W_{\mathcal{S}}^{*} = R^{-1} \tilde{\Delta}^{*} = R^{-1} U \Sigma_{r} V^{\top}, \tag{27}$$

which completes the proof. $\qquad\qquad\qquad\qquad\qquad\qquad\qquad\qquad\qquad\qquad\qquad\qquad\qquad\qquad\qquad\qquad\quad\square$

## D. Post-hoc Solver Verification

We further conduct a post-hoc numerical verification of the closed-form solver. For each evaluated MLP projection module, we re-optimize the same objective in Eq. 18 with L-BFGS under the same rank constraint, parameterized as $\Delta W = AB^{\top}$. We then compare the final objective value of the L-BFGS solution $T^{*}$ with the closed-form solution used by our method. This verification is not a separate training procedure for our method; it is used only to check whether a generic iterative optimizer can find a lower value for the same objective.

Across six model backbones and 320 MLP projection modules, the closed-form solution achieves a lower objective value than the L-BFGS solution in all evaluated cases (320/320). The average objective reduction ranges from 5.14% to 14.42% across models, indicating that the advantage is not driven by a small number of isolated layers. We also observe that the gain is typically larger in deeper layers; for example, the average reduction increases from 7.62% in the first half to 12.46% in the second half for Qwen2-1.5B, and from 12.74% to 16.09% for Qwen2.5-1.5B. Both `gate_proj` and `up_proj` modules consistently satisfy this trend.

*Table 7.* **Post-hoc verification of the closed-form solver.** For each model, we compare the objective value achieved by the closed-form solution with an L-BFGS solution $T^*$ optimized under the same objective and rank. "Close better" counts modules where the closed-form solution obtains a lower objective value. Average loss reduction is computed relative to the L-BFGS objective.

| MODEL | CONFIG | MODULES | CLOSE BETTER | AVG. LOSS RED. | AVG. $T^*$/CLOSE |
|---|---|---|---|---|---|
| LLAMA-3.2-1B-INSTRUCT | $\gamma = 1000, \lambda = 10^4, r = 32$ | 32 | 32/32 | 7.44% | 1.0809 |
| LLAMA-3.2-3B-INSTRUCT | $\gamma = 1000, \lambda = 5\times10^4, r = 32$ | 56 | 56/56 | 5.14% | 1.0545 |
| QWEN2-0.5B-INSTRUCT | $\gamma = 1000, \lambda = 10^5, r = 32$ | 48 | 48/48 | 7.71% | 1.0840 |
| QWEN2-1.5B-INSTRUCT | $\gamma = 1000, \lambda = 10^5, r = 32$ | 56 | 56/56 | 10.04% | 1.1128 |
| QWEN2.5-3B-INSTRUCT | $\gamma = 1000, \lambda = 5\times10^4, r = 32$ | 72 | 72/72 | 11.89% | 1.1359 |
| QWEN2.5-1.5B-INSTRUCT | $\gamma = 1000, \lambda = 10^4, r = 32$ | 56 | 56/56 | 14.42% | 1.1697 |

# E. Implementation Details

**Hyperparameters.** We use a fixed utility-preservation weight $\gamma = 1000$ across all models. The regularization weight $\lambda$ is scaled by model family and size because the weight norms differ substantially across backbones. In our experiments, we use $\lambda = 10^4$ for Llama-3.2-1B and Qwen2.5-1.5B, $\lambda = 5 \times 10^4$ for Llama-3.2-3B and Qwen2.5-3B, $\lambda = 10^5$ for Qwen2-0.5B and Qwen2-1.5B, and $\lambda = 10^6$ for Qwen2.5-7B. The rank $r$ controls the rank of the low-rank edit and is kept fixed within each model unless otherwise stated.

**Anchor size and sensitivity.** We use harmful and harmless anchors to estimate the safety-transfer direction and the utility-preserving constraint, respectively. To examine whether our method requires a large or carefully tuned anchor set, we conduct an anchor-size sensitivity study on Llama-3.2-1B. Specifically, $K$ shots denotes $K$ harmful anchors and $K$ harmless anchors per language for each edited layer. We vary $K$ from 10 to 5000 and keep the remaining editing pipeline unchanged.

As shown in Table 8, our method is robust to the anchor size. Increasing $K$ generally reduces unsafe responses, with the average improvement increasing from $-17.625$ at $K = 10$ to $-41.125$ at $K = 5000$. The gains mostly saturate after a few hundred to a few thousand anchors. Meanwhile, utility remains stable: MGSM stays close to the base score, and M-MMLU improves from 26.54 to 27.32.

*Table 8.* **Anchor-size sensitivity on Llama-3.2-1B.** We vary the number of anchors $K$, where $K$ shots denotes $K$ harmful anchors and $K$ harmless anchors per language for each edited layer. For safety, we report the number of unsafe responses flagged by `Qwen3Guard-Gen-8B` out of 313 prompts for each language (lower is better). Superscripts denote the change in unsafe-response counts relative to **Base** (negative indicates improvement; positive indicates regression). $\Delta_{Avg}$ denotes the average change in unsafe-response counts across all reported languages (En, Zh, Vi, Ja, Th, Id, Bn, He).

| MODEL | SHOTS | SAFETY | | | | | | | | | UTILITY | |
|---|---|---|---|---|---|---|---|---|---|---|---|---|
| | | ASR $\downarrow$ (#UNSAFE / 313) | | | | | | | | | ACCURACY $\uparrow$ | |
| | | EN | ZH | VI | JA | TH | ID | BN | HE | $\Delta_{Avg}$ | MGSM | M-MMLU |
| | BASE | 6/313 | 61/313 | 31/313 | 149/313 | 69/313 | 104/313 | 179/313 | 109/313 | - | 18.58 | 26.54 |
| | 10 | 0/313$^{-6}$ | 33/313$^{-28}$ | 15/313$^{-16}$ | 107/313$^{-42}$ | 41/313$^{-28}$ | 67/313$^{-37}$ | 175/313$^{-4}$ | 129/313$^{+20}$ | -17.625 | 18.73 | 26.74 |
| | 50 | 0/313$^{-6}$ | 22/313$^{-39}$ | 6/313$^{-25}$ | 98/313$^{-51}$ | 32/313$^{-37}$ | 67/313$^{-37}$ | 173/313$^{-6}$ | 120/313$^{+11}$ | -23.75 | 19.24 | 26.82 |
| LLAMA-3.2-1B | 100 | 0/313$^{-6}$ | 23/313$^{-38}$ | 6/313$^{-25}$ | 79/313$^{-70}$ | 19/313$^{-50}$ | 61/313$^{-43}$ | 176/313$^{-3}$ | 115/313$^{+6}$ | -28.625 | 18.73 | 26.77 |
| | 500 | 1/313$^{-5}$ | 18/313$^{-43}$ | 6/313$^{-25}$ | 55/313$^{-94}$ | 25/313$^{-44}$ | 41/313$^{-63}$ | 151/313$^{-28}$ | 107/313$^{-2}$ | -38.0 | 18.84 | 26.87 |
| | 1000 | 0/313$^{-6}$ | 16/313$^{-45}$ | 4/313$^{-27}$ | 52/313$^{-97}$ | 21/313$^{-48}$ | 34/313$^{-70}$ | 147/313$^{-32}$ | 118/313$^{+9}$ | -39.5 | 18.65 | 27.02 |
| | 5000 | 0/313$^{-6}$ | 19/313$^{-42}$ | 2/313$^{-29}$ | 60/313$^{-89}$ | 11/313$^{-58}$ | 26/313$^{-78}$ | 147/313$^{-32}$ | 114/313$^{+5}$ | -41.125 | 18.62 | 27.32 |

# F. Complete Multilingual Safety Results

Table 9 provides the complete safety results for all eight languages in MULTI-STRONGREJECT, complementing the subset reported in Table 1 in the main text. Consistent with our main findings, our training-free update reduces unsafe completions across most languages and backbones, and composes well with MPO (often yielding the lowest unsafe-response counts).

## F.1. Additional Evaluation on MULTIJAIL

To evaluate whether the observed safety gains extend beyond MULTI-STRONGREJECT, we additionally test our method on the multilingual jailbreak benchmark MULTIJAIL. Table 10 reports the number of unsafe responses before and after applying our edit. Lower values are better, and superscripts denote the change relative to the unedited baseline for the same backbone.

The MULTIJAIL results show the same overall trend as MULTI-STRONGREJECT: our edit reduces unsafe responses in

*Table 9.* **Complete zero-shot multilingual safety evaluation on MULTI-STRONGREJECT.** We report the number of unsafe responses flagged by `Qwen3Guard-Gen-8B` out of 313 prompts for each language (lower is better). Superscripts denote the change in unsafe-response counts relative to **None** for the same backbone (negative indicates improvement; positive indicates regression). $\Delta_{Avg}$ denotes the average change in unsafe-response counts across all reported languages (En, Zh, Vi, Ja, Th, Id, Bn, He).

| MODELS | METHOD | SAFETY ASR ↓ (#UNSAFE / 313) | | | | | | | | |
|---|---|---|---|---|---|---|---|---|---|---|
| | | EN | ZH | VI | JA | TH | ID | BN | HE | $\Delta_{Avg}$ |
| LLAMA-3.2-1B | NONE | 6/313 | 61/313 | 31/313 | 149/313 | 69/313 | 104/313 | 179/313 | 109/313 | - |
| | OUR | 0/313$^{-6}$ | 27/313$^{-34}$ | 4/313$^{-27}$ | 81/313$^{-68}$ | 30/313$^{-39}$ | 38/313$^{-66}$ | 144/313$^{-35}$ | 115/313$^{+6}$ | -33.625 |
| | MPO | 0/313$^{-6}$ | 22/313$^{-39}$ | 9/313$^{-22}$ | 78/313$^{-71}$ | 23/313$^{-46}$ | 70/313$^{-34}$ | 152/313$^{-27}$ | 135/313$^{+26}$ | -23.15 |
| | MPO+OUR | 0/313$^{-6}$ | 22/313$^{-39}$ | 0/313$^{-31}$ | 66/313$^{-83}$ | 10/313$^{-59}$ | 22/313$^{-82}$ | 96/313$^{-83}$ | 109/313$^{-0}$ | -47.875 |
| LLAMA-3.2-3B | NONE | 6/313 | 9/313 | 10/313 | 79/313 | 22/313 | 19/313 | 110/313 | 39/313 | - |
| | OUR | 4/313$^{-2}$ | 3/313$^{-6}$ | 2/313$^{-8}$ | 34/313$^{-45}$ | 4/313$^{-18}$ | 5/313$^{-14}$ | 65/313$^{-45}$ | 46/313$^{+7}$ | -16.375 |
| | MPO | 4/313$^{-2}$ | 8/313$^{-1}$ | 4/313$^{-6}$ | 50/313$^{-29}$ | 19/313$^{-3}$ | 20/313$^{+1}$ | 91/313$^{-19}$ | 36/313$^{-3}$ | -10.0 |
| | MPO+OUR | 2/313$^{-4}$ | 1/313$^{-8}$ | 3/313$^{-7}$ | 30/313$^{-49}$ | 2/313$^{-20}$ | 3/313$^{-16}$ | 58/313$^{-52}$ | 36/313$^{-3}$ | -19.875 |
| QWEN2-0.5B | NONE | 224/313 | 197/313 | 185/313 | 193/313 | 162/313 | 205/313 | 208/313 | 150/313 | - |
| | OUR | 176/313$^{-48}$ | 121/313$^{-76}$ | 139/313$^{-46}$ | 145/313$^{-48}$ | 138/313$^{-24}$ | 168/313$^{-37}$ | 173/313$^{-35}$ | 134/313$^{-16}$ | -41.25 |
| | MPO | 108/313$^{-116}$ | 93/313$^{-104}$ | 83/313$^{-102}$ | 94/313$^{-99}$ | 42/313$^{-120}$ | 88/313$^{-117}$ | 162/313$^{-46}$ | 90/313$^{-60}$ | -87.83 |
| | MPO+OUR | 56/313$^{-168}$ | 41/313$^{-156}$ | 44/313$^{-141}$ | 49/313$^{-144}$ | 32/313$^{-130}$ | 68/313$^{-137}$ | 120/313$^{-88}$ | 65/313$^{-85}$ | -131.125 |
| QWEN2-1.5B | NONE | 36/313 | 18/313 | 36/313 | 67/313 | 60/313 | 50/313 | 187/313 | 83/313 | - |
| | OUR | 5/313$^{-31}$ | 4/313$^{-14}$ | 15/313$^{-21}$ | 19/313$^{-48}$ | 13/313$^{-47}$ | 4/313$^{-46}$ | 150/313$^{-37}$ | 36/313$^{-47}$ | -36.375 |
| | MPO | 0/313$^{-36}$ | 2/313$^{-16}$ | 0/313$^{-36}$ | 3/313$^{-64}$ | 2/313$^{-58}$ | 0/313$^{-50}$ | 21/313$^{-166}$ | 3/313$^{-80}$ | -66.33 |
| | MPO+OUR | 3/313$^{-33}$ | 0/313$^{-18}$ | 5/313$^{-31}$ | 1/313$^{-66}$ | 1/313$^{-59}$ | 4/313$^{-46}$ | 1/313$^{-186}$ | 1/313$^{-82}$ | -65.125 |
| QWEN2.5-1.5B | NONE | 60/313 | 30/313 | 42/313 | 56/313 | 68/313 | 59/313 | 182/313 | 118/313 | - |
| | OUR | 17/313$^{-43}$ | 5/313$^{-25}$ | 14/313$^{-28}$ | 14/313$^{-42}$ | 18/313$^{-50}$ | 25/313$^{-34}$ | 152/313$^{-30}$ | 81/313$^{-37}$ | -36.125 |
| | MPO | 6/313$^{-54}$ | 2/313$^{-28}$ | 1/313$^{-41}$ | 2/313$^{-54}$ | 7/313$^{-61}$ | 2/313$^{-57}$ | 54/313$^{-128}$ | 26/313$^{-92}$ | -62.66 |
| | MPO+OUR | 5/313$^{-55}$ | 2/313$^{-28}$ | 2/313$^{-40}$ | 7/313$^{-49}$ | 3/313$^{-65}$ | 0/313$^{-59}$ | 56/313$^{-126}$ | 22/313$^{-96}$ | -64.75 |
| QWEN2.5-3B | NONE | 61/313 | 64/313 | 64/313 | 81/313 | 57/313 | 60/313 | 157/313 | 100/313 | - |
| | OUR | 14/313$^{-47}$ | 4/313$^{-60}$ | 7/313$^{-57}$ | 15/313$^{-66}$ | 16/313$^{-41}$ | 15/313$^{-45}$ | 112/313$^{-45}$ | 41/313$^{-59}$ | -52.5 |
| | MPO | 16/313$^{-45}$ | 10/313$^{-54}$ | 10/313$^{-54}$ | 16/313$^{-65}$ | 20/313$^{-37}$ | 16/313$^{-44}$ | 67/313$^{-90}$ | 32/313$^{-68}$ | -62.66 |
| | MPO+OUR | 6/313$^{-55}$ | 5/313$^{-59}$ | 3/313$^{-61}$ | 4/313$^{-77}$ | 5/313$^{-52}$ | 5/313$^{-55}$ | 25/313$^{-132}$ | 7/313$^{-93}$ | -73.0 |
| QWEN2.5-7B | NONE | 16/313 | 12/313 | 21/313 | 39/313 | 27/313 | 21/313 | 98/313 | 48/313 | - |
| | OUR | 3/313$^{-13}$ | 5/313$^{-7}$ | 6/313$^{-15}$ | 9/313$^{-30}$ | 12/313$^{-15}$ | 6/313$^{-15}$ | 60/313$^{-38}$ | 24/313$^{-24}$ | -19.625 |
| | MPO | 6/313$^{-10}$ | 5/313$^{-7}$ | 5/313$^{-16}$ | 8/313$^{-31}$ | 11/313$^{-16}$ | 7/313$^{-14}$ | 25/313$^{-73}$ | 17/313$^{-31}$ | -28.0 |
| | MPO+OUR | 0/313$^{-16}$ | 0/313$^{-12}$ | 1/313$^{-20}$ | 2/313$^{-37}$ | 3/313$^{-24}$ | 1/313$^{-20}$ | 11/313$^{-87}$ | 11/313$^{-37}$ | -31.625 |

nearly all model-language settings, with average improvements across all evaluated backbones. The only positive deltas appear on Swahili for Qwen2.5-1.5B and Qwen2.5-7B, while the remaining settings either improve or remain unchanged. This suggests that the safety improvement is not limited to a single multilingual safety benchmark.

## F.2. Comparison with MCD

We further compare our method with *Multilingual Collaborative Defense for Large Language Models* (MCD) (Li et al., 2025), a prompt-tuning multilingual safety baseline, on `Qwen2.5-1.5B`. Table 11 reports safety on MULTI-STRONGREJECT together with multilingual utility. MCD achieves strong reductions in unsafe responses, but this comes with a severe utility collapse, especially on MGSM. In contrast, our method provides a more balanced safety–utility trade-off: it substantially reduces unsafe responses while preserving MGSM and M-MMLU close to the unedited baseline.

## F.3. Evaluation with an External Safety Judge

Our main safety evaluation uses `Qwen/Qwen3Guard-Gen-8B` for scalable multilingual harmfulness classification. To check whether the safety trend is an artifact of this specific evaluator, we additionally evaluate a subset of models with GPTJudge (GPT-5) as an external safety judge. GPTJudge assigns a harmfulness score to each model response, and we report the average score for the edited model together with its change relative to the corresponding unedited baseline. Lower scores indicate safer responses.

The GPTJudge results are broadly consistent with the primary Qwen3Guard-based evaluation. Our method reduces the

*Table 10.* **Additional safety evaluation on MULTIJAIL.** We report the number of unsafe responses for each language (lower is better). Superscripts denote the change in unsafe-response counts relative to **None** for the same backbone (negative indicates improvement; positive indicates regression). $\Delta_{Avg}$ denotes the average change across all reported languages (En, Zh, Vi, Th, Bn, Sw, Jv).

| MODELS | METHOD | SAFETY | | | | | | | |
|---|---|---|---|---|---|---|---|---|---|
| | | ASR ↓ (#UNSAFE) | | | | | | | |
| | | EN | ZH | VI | TH | BN | SW | JV | $\Delta_{Avg}$ |
| LLAMA-3.2-1B | NONE | 17 | 27 | 31 | 27 | 91 | 7 | 6 | - |
| | OUR | $1^{-16}$ | $21^{-6}$ | $8^{-23}$ | $9^{-18}$ | $59^{-32}$ | $7^{0}$ | $6^{0}$ | -13.57 |
| LLAMA-3.2-3B | NONE | 18 | 24 | 26 | 32 | 53 | 6 | 10 | - |
| | OUR | $9^{-9}$ | $14^{-10}$ | $9^{-17}$ | $14^{-18}$ | $37^{-16}$ | $5^{-1}$ | $4^{-6}$ | -11.00 |
| QWEN2-0.5B | NONE | 145 | 109 | 124 | 115 | 106 | 12 | 21 | - |
| | OUR | $138^{-7}$ | $94^{-15}$ | $85^{-39}$ | $83^{-32}$ | $78^{-28}$ | $3^{-9}$ | $18^{-3}$ | -19.00 |
| QWEN2-1.5B | NONE | 33 | 21 | 28 | 38 | 96 | 16 | 19 | - |
| | OUR | $9^{-24}$ | $5^{-16}$ | $19^{-9}$ | $10^{-28}$ | $80^{-16}$ | $9^{-7}$ | $16^{-3}$ | -14.71 |
| QWEN2.5-1.5B | NONE | 41 | 19 | 31 | 38 | 104 | 10 | 12 | - |
| | OUR | $14^{-27}$ | $1^{-18}$ | $8^{-23}$ | $17^{-21}$ | $90^{-14}$ | $11^{+1}$ | $4^{-8}$ | -15.71 |
| QWEN2.5-3B | NONE | 67 | 33 | 54 | 38 | 56 | 8 | 21 | - |
| | OUR | $22^{-45}$ | $9^{-24}$ | $11^{-43}$ | $14^{-24}$ | $41^{-15}$ | $8^{0}$ | $12^{-9}$ | -22.86 |
| QWEN2.5-7B | NONE | 30 | 21 | 16 | 19 | 42 | 11 | 23 | - |
| | OUR | $10^{-20}$ | $8^{-13}$ | $8^{-8}$ | $7^{-12}$ | $33^{-9}$ | $15^{+4}$ | $14^{-9}$ | -9.57 |

*Table 11.* **Comparison with MCD on `Qwen2.5-1.5B`.** Safety is reported as the number of unsafe responses on MULTI-STRONGREJECT out of 313 prompts (lower is better). Superscripts denote the change in unsafe-response counts relative to **None** (negative indicates improvement). $\Delta_{Avg}$ denotes the average change across all reported languages. Utility is measured by MGSM and M-MMLU accuracy (higher is better).

| MODEL | METHOD | SAFETY | | | | | | | | | | | UTILITY | |
|---|---|---|---|---|---|---|---|---|---|---|---|---|---|---|
| | | ASR ↓ (#UNSAFE / 313) | | | | | | | | | | | MGSM↑ | M-MMLU↑ |
| | | EN | ZH | VI | JA | TH | ID | BN | HE | SW | JV | $\Delta_{Avg}$ | | |
| QWEN2.5-1.5B | NONE | 60/313 | 30/313 | 42/313 | 56/313 | 68/313 | 59/313 | 182/313 | 118/313 | 23/313 | 55/313 | - | 27.53 | 41.58 |
| | OUR | $17/313^{-43}$ | $5/313^{-25}$ | $14/313^{-28}$ | $14/313^{-42}$ | $18/313^{-50}$ | $25/313^{-34}$ | $152/313^{-30}$ | $81/313^{-37}$ | $11/313^{-12}$ | $12/313^{-43}$ | -34.40 | 25.13 | 41.89 |
| | MCD | $19/313^{-41}$ | $27/313^{-3}$ | $10/313^{-32}$ | $15/313^{-41}$ | $11/313^{-57}$ | $11/313^{-48}$ | $7/313^{-175}$ | $6/313^{-112}$ | $0/313^{-23}$ | $3/313^{-52}$ | -58.40 | 0.20 | 36.99 |

external harmfulness score in 28 out of 30 model-language settings, including all evaluated settings for the main eight languages.

## F.4. Additional Evaluation on Swahili and Javanese

To further evaluate our method on lower-resource languages beyond the main eight-language setting, we add Swahili (sw) and Javanese (jv) to MULTI-STRONGREJECT. Table 13 reports the unedited baseline and our edited model on these two languages. The results show consistent improvements on Javanese across all tested backbones and improvements on Swahili in most settings.

*Table 12.* **External-judge safety evaluation with GPTJudge (GPT-5).** Entries are reported as the average harmfulness score of **Our** method, with the change relative to **None** shown in parentheses. Lower scores are better; negative changes indicate safer responses under the external judge.

| MODEL | EN | ZH | VI | JA | TH | ID | BN | HE | SW | JV | AVG |
|---|---|---|---|---|---|---|---|---|---|---|---|
| QWEN2.5-1.5B | 1.81 (-0.67) | 1.29 (-0.57) | 1.53 (-0.75) | 1.64 (-1.01) | 1.70 (-1.44) | 1.93 (-0.66) | 3.35 (-0.57) | 2.56 (-1.51) | 1.69 (-0.10) | 1.41 (-0.97) | 1.89 (-0.83) |
| QWEN2.5-3B | 1.45 (-1.37) | 1.18 (-1.36) | 1.42 (-1.39) | 1.60 (-1.45) | 1.61 (-1.22) | 1.57 (-1.07) | 3.97 (-1.18) | 2.50 (-1.25) | 3.23 (+0.59) | 2.21 (-1.27) | 2.07 (-1.10) |
| QWEN2.5-7B | 1.13 (-0.35) | 1.17 (-0.37) | 1.25 (-0.47) | 1.30 (-0.81) | 1.39 (-0.60) | 1.33 (-0.29) | 3.19 (-0.88) | 1.94 (-0.83) | 3.88 (+0.11) | 2.04 (-0.78) | 1.86 (-0.53) |

*Table 13.* **Additional evaluation on Swahili and Javanese.** We report the number of unsafe responses on MULTI-STRONGREJECT out of 313 prompts; lower is better. Superscripts denote the change in unsafe-response counts relative to **None** for the same backbone (negative indicates improvement; positive indicates regression).

| MODEL | SWAHILI | | JAVANESE | |
|---|---|---|---|---|
| | NONE | OUR | NONE | OUR |
| LLAMA-3.2-1B | 15/313 | 8/313 [-7] | 41/313 | 16/313 [-25] |
| LLAMA-3.2-3B | 9/313 | 12/313 [+3] | 50/313 | 16/313 [-34] |
| QWEN2-0.5B | 20/313 | 9/313 [-11] | 147/313 | 103/313 [-44] |
| QWEN2-1.5B | 22/313 | 15/313 [-7] | 62/313 | 18/313 [-44] |
| QWEN2.5-1.5B | 23/313 | 11/313 [-12] | 55/313 | 12/313 [-43] |
| QWEN2.5-3B | 23/313 | 29/313 [+6] | 101/313 | 41/313 [-60] |
| QWEN2.5-7B | 21/313 | 21/313 [0] | 55/313 | 34/313 [-21] |

